# Perinatal Outcomes and Neurodevelopment 1 Year after Birth in Discordant Twins According to Chorionicity

**DOI:** 10.3390/medicina59030493

**Published:** 2023-03-02

**Authors:** Mi Ju Kim, Hyun Mi Kim, Hyun-Hwa Cha, Haemin Kim, Hyo-Shin Kim, Bong Seon Lim, Won Joon Seong

**Affiliations:** 1Department of Obstetrics and Gynecology, Kyungpook National University Hospital, School of Medicine, Kyungpook National University, Daegu 41944, Republic of Korea; 2High Risk Maternal Fetus Intensive Care Center, Kyungpook National University Chilgok Hospital, Daegu 41404, Republic of Korea

**Keywords:** chorionicity, maternal complications, neonatal outcomes, neurodevelopment, twin weight discordancy

## Abstract

*Background and Objectives*: This study aimed to compare maternal complications, perinatal outcomes, and neurodevelopment 1 year after the birth between concordant and discordant twins in monochorionic and dichorionic twins. *Materials and Methods*: This retrospective study included twin pregnancies delivered between 24 + 1 and 38 + 2 weeks of gestation between January 2011 and September 2019. Chorionicity was confirmed by ultrasonography and was categorized into monochorionic and dichorionic. Each was then divided into two groups (concordant and discordant) according to birth weight discordancy. Maternal complications and neonatal outcomes, including neurodevelopmental delays, were compared between the two groups. *Results*: A total of 298 pairs of twin pregnancies were enrolled, of which 58 (19.26%) women were pregnant with monochorionic diamniotic twins and 240 (80.54%) with dichorionic diamniotic twins. In both monochorionic and dichorionic twins, the discordant twins had a greater incidence of emergency deliveries because of iatrogenic causes than the concordant twins. Among dichorionic twins, discordant twins had lower birth weight rates and higher hospitalization rates and morbidities than concordant twins. Among monochorionic twins, discordant twins had a lower birth weight and higher neonatal mortality than concordant twins. The neonatal size was not a predictor of neurodevelopment in this group. Based on the logistic regression analysis, male sex, respiratory distress syndrome, and bronchopulmonary dysplasia were risk factors for the neurodevelopmental delay; birth weight discordancy was significant only in dichorionic twins. *Conclusions*: Perinatal outcomes in discordant twins may be poor, and neurodevelopment 1 year after birth was worse in discordant twins than in concordant twins. Discordancy in twins can be a risk factor for neurodevelopmental delay.

## 1. Introduction

Approximately 1–3% of all pregnancies are multiple pregnancies. As the use of artificial reproductive techniques (ART) continues to increase with advanced maternal age, the number of twin pregnancies is also increasing. The rate of twin pregnancies increased by 76% (from 18.9 to 33.3 per 1000 births) from 1980 to 2009 [1]. However, twin pregnancies can result in several obstetrical problems, such as preterm birth (PTB), preeclampsia, gestational diabetes mellitus (GDM), and operative delivery. This is also associated with neonatal complications, such as respiratory difficulties caused by PTB, fetal growth restriction (FGR), and congenital anomalies [2,3,4,5]. In twin pregnancy, weight discordancy is considered important for neonatal outcomes. The birth weight of two babies may differ even in the absence of twin-specific complications such as twin-to-twin transfusion syndrome (TTTS). Several studies have reported on how weight discordancy affects neonatal prognosis [6]. According to the American College of Obstetricians and Gynecologists (ACOG) and the International Society of Obstetrics and Gynecology (ISUOG) [7], weight discordancy is defined as a birth weight difference of >20% between two babies. Twin discordancy is a twin-specific condition with a total incidence of 10–15%, and has a consistent prevalence regardless of chorionicity [8,9]. Unfortunately, twin discordancy is also closely related to FGR, which negatively affects neonatal outcomes [10]. However, the risk of neonatal morbidity and mortality was reported to be increased in discordant twins despite the absence of FGR [11]. Twin weight discordancy is related to poor neonatal outcomes regardless of chorionicity; however, a study reported that monochorionic (MC) twin discordancy had a worse prognosis at all levels of discordance [10].

In discordant twins, chorionicity is an independent factor for neonatal outcomes [12,13]. Therefore, this study classified twins according to chorionicity (MC vs. dichorionic [DC]) and compared the neonatal outcomes of discordant and concordant twins from each group. We also compared the neonatal outcomes of large and small discordant babies with their concordant counterparts to determine the relationship between discordant twin size and prognosis.

The primary aim of this study was to compare the maternal outcomes of discordant and concordant twins according to chorionicity and compare the neonatal outcomes of large and small discordant twins with those of concordant twins. The secondary aim was to determine the effect of discordancy on neonatal neurodevelopment 1 year after birth.

## 2. Materials and Methods

In this study, we retrospectively reviewed the medical records of 350 women with twin pregnancies who delivered between 24 + 1 and 38 + 2 weeks of gestation at Chilgok Kyungpook National University Hospital in Daegu, South Korea, between January 2011 and September 2019. Figure 1 shows a flow chart of enrollment. Among them, three pregnancies of MC monoamniotic twin pregnancy and four cases where chorionicity could not be confirmed (unknown chorionicity) were excluded. Chorionicity was determined via ultrasonography in early pregnancy and, if unclear, further confirmed using placental biopsy after delivery. Moreover, 17 cases of a single fetal demise in utero, 10 cases of suspected TTTS, and 18 cases with major chromosomal anomalies related to morbidity and mortality were excluded. A total of 298 sets of twins were studied.

Weight discordancy was calculated using the following formula: (birth weight of the larger twin—birth weight of the smaller twin) × 100/birth weight of the larger twin. According to the consensus of ACOG and ISUOG, values >20% define cases of discordant twins, whereas those <20% define cases of concordant twins [7]. Discordant and concordant twins were also classified according to chorionicity (MC and DC) and assessed for their maternal complications and neonatal outcomes. The following maternal characteristics were assessed: age at delivery, parity, pre-pregnancy body mass index (BMI), and use of ARTs. Ovulation induction, intrauterine insemination, in vitro fertilization, and embryo transfer were all considered ARTs. Pregnancy-related complications, such as preeclampsia, placenta previa, GDM, threatened preterm labor, and postpartum hemorrhage (PPH), were also evaluated. Preeclampsia was defined by high blood pressure (at least 140/90 mmHg) and proteinuria (at least 300 mg for 24 h) after 20 weeks of gestation [14]. Between 24 and 28 weeks of gestation, a 50-g glucose tolerance test was used to screen for GDM, and a 100-g glucose tolerance test was used as a diagnostic test in accordance with the Carpenter–Coustan criteria [15]. Placenta previa includes placenta previa totalis and low-lying placenta, which were classified based on surgical findings. PPH was defined as a case of suspected bleeding of ≥1000 mL before and after delivery or progressive anemia requiring blood transfusion. In terms of the reason for delivery, elective deliveries were defined as those on a scheduled date without an event; spontaneous deliveries were those due to labor or amniotic membrane rupture; and iatrogenic deliveries were those medically indicated for the wellbeing of the mother and fetus. The following neonatal outcomes were also analyzed: gestational age at delivery, sex, neonatal birthweight, 1-min and 5-min Apgar scores, neonatal intensive care unit (NICU) admission, neonatal composite morbidity, and mortality. Composite morbidity was defined as the presence of any of the following in a neonate: respiratory distress syndrome (RDS), bronchopulmonary dysplasia, intraventricular hemorrhage (IVH), periventricular leukomalacia (PVL), necrotizing enterocolitis (NEC), and sepsis. With the concordant twin group as a control, the neonatal outcomes of the discordant large and small babies were investigated. In discordant twins, the outcomes between large and small babies were also compared. We reviewed the rate of neurodevelopmental delay in cases with fine and gross motor disturbances observed through physical examination and assessed using the Bayley scale and in those that required rehabilitation 1 year after birth [16]. Scores of <70 points were considered underdeveloped, and frequencies were measured between the groups. Since most twins were born prematurely at our center, they were regularly followed up by a pediatrician, regardless of discordancy. In particular, if neurodevelopmental delay was suspected, follow-up examinations were performed every 6 months and 1 year at the Department of Pediatric Rehabilitation Medicine. All procedures were performed in accordance with relevant guidelines and regulations.

### 2.1. Statistical Analysis

All data were analyzed using IBM SPSS version 26.0 (IBM Corp., Armonk, NY, USA) and R version 4.0.0 (Vienna, Austria; www.r-project.org (accessed on 24 April 2020)). Data were compared using the Mann–Whitney U-test for continuous numerical data and the chi-square test for binary categorical data. After performing a normality test, the mean ± standard deviation was used to present continuous variables with normal distributions, whereas numbers (percentage) were used for binary categorical data. Variables with a *p*-value < 0.05 based on the Mann–Whitney U-test or chi-square test were used. A logistic regression analysis was conducted to identify the independent variables predictive of developmental delay according to chorionicity. Odds ratios and 95% confidence intervals (CIs) were calculated, and *p*-values of <0.05 were considered statistically significant.

### 2.2. Ethical Approval

This study was approved by the Institutional Review Board (IRB) of Chilgok Kyungpook National University Hospital (IRB No. KNUCH 2020-06-015, 2020.7.2). Informed consent was not obtained from the study participants because of the retrospective nature of the medical record review. The informed consent waiver was approved by the IRB of Chilgok Kyungpook National University Hospital.

## 3. Results

In total, 298 sets of twins were included in this study. Of these, 58 (19.46%) sets of twins were MC diamniotic (MCDA) twins, and 240 (80.54%) sets of twins were DC diamniotic (DCDA) twins. Among MCDA twins, 47 (81.03%) sets of twins were concordant, and 11 (18.97%) were discordant, whereas among DCDA twins, 190 (79.17%) sets of twins were concordant, and 50 (20.83%) were discordant.

Table 1 illustrates the maternal characteristics and obstetrical complications between the concordant and discordant twins according to chorionicity. In MC twins, no significant differences were observed in maternal parity, pre-pregnancy BMI, ART use, the incidence of preeclampsia or GDM, or premature rupture of the membrane, regardless of discordancy. However, MC discordant twins tended to have a higher maternal age at delivery (30.28 ± 3.75 vs. 32.55 ± 3.58, *p* = 0.011), a higher frequency of placenta previa (0.0% vs. 9.01%, *p* = 0.041), and a higher incidence of PPH (0.0% vs. 9.1%, *p* = 0.041) than concordant twins. In DC twins, no significant differences were observed in maternal age, placenta previa, PPH, or threatened preterm labor, regardless of discordancy. However, more nulliparous women (67.9% vs. 78.0%, *p* = 0.028), higher frequency of ART (60.9% vs. 74.0%, *p* = 0.028), and a higher incidence of preeclampsia (8.5% vs. 20.00%, *p* = 0.02) were observed in DC discordant twins than in concordant twins. Overall, concordant twins had a higher frequency of elective delivery than discordant twins (44.3% vs. 18.5%, *p* < 0.001), and delivery due to iatrogenic reasons was relatively low (9.1% vs. 29.2%, *p* < 0.001) in both MC and DC twins.

Table 2 compares the neonatal outcomes between the concordant and discordant twins according to chorionicity. No statistically significant differences were observed in the gestational age at delivery, male sex, and 1- and 5-min Apgar scores of ≤7 among MC and DC twins when grouped according to discordancy. MC discordant twins had lower birth weight (1802.50 ± 779.22 vs. 2194.89 ± 471.05, *p* = 0.032) and higher neonatal mortality (0.00% vs. 13.64%, *p* = 0.004) than MC concordant twins. However, they did not differ in terms of the frequency of NICU admission and the composite morbidity of RDS, IVH, PVL, NEC, and sepsis. On the contrary, DC discordant twins had lower birth weight (2038.60 ± 553.76 vs. 2258.94 ± 479.99, *p* < 0.001), higher frequency of NICU admission (89.0% vs. 65.4%, *p* < 0.001), higher composite neonatal morbidity (38.0% vs. 26.8%, *p* = 0.039), and a more frequent neurodevelopmental delay at 1 year after birth (2.9% vs. 13.0%, *p* < 0.001) than DC concordant twins.

Table 3 compares the neonatal outcomes of concordant twins with both discordant large neonates and discordant small neonates according to chorionicity. The smaller counterparts of each discordant twin were grouped into one and were compared with concordant babies. Likewise, the larger counterparts of each discordant twin were grouped into one and were compared with concordant babies. Then, the smaller and larger twin groups were compared. In MC twins, no difference was observed in the overall neonatal outcomes between discordant large and small neonates. Even when comparing concordant and discordant large neonates, no significant differences were observed in birth weight, respiratory complications, or developmental delay. However, discordant small neonates had a lower birth weight (2194.89 ± 471.05 vs. 1517.27 ± 610.48, *p* < 0.001) and higher incidence of neonatal death (0% vs. 18.2%, *p* = 0.003) than concordant neonates. Notably, when comparing concordant neonates and discordant large neonates among DC twins, significant differences were observed in PVL incidence (1.1% vs. 10.0%, *p* < 0.001) and neurodevelopmental delay (2.9% vs. 18.0%, *p* < 0.001). Discordant large neonates were more prevalent among males (72.0% vs. 30.0%, *p* < 0.001) than discordant small babies. The 1- and 5-min Apgar scores of ≤7 and the incidence of neonatal death were not statistically significant between the two groups; however, NICU admission was higher in small neonates (100.0% vs. 78.0%, *p* < 0.001).

Table 4 shows the neonatal characteristics of MC and DC twins, comparing those with and without developmental delays. Among DC twins, those with developmental delay had an earlier delivery (32.63 ± 3.42 vs. 35.16 ± 2.35 weeks, *p* = 0.001) and a lower birth weight (1876.67 ± 590.75 vs. 2230.74 ± 493.11, *p* = 0.001) than those without. Furthermore, male sex was common (75.0% vs. 51.1%, *p* = 0.038), and the NICU hospitalization frequency was also high (91.67% vs. 69.08%, *p* = 0.033) in this subgroup. The presence of placental infection, such as chorioamnionitis, was higher in the group with neurodevelopmental delay, but this difference was not statistically significant (12.5% vs. 6.4%, *p* = 0.620). Among those with neurodevelopmental delays, discordant twins (54.2% vs. 19.1%, *p* < 0.001) and composite morbidity (66.7% vs. 27.2%, *p* < 0.001) were more common. MC twins with neurodevelopmental delay had a lower birth weight (1565.00 ± 652.04 vs. 2167.22 ± 529.50, *p* = 0.002) than those without. No statistically significant differences were observed in NICU admission, composite morbidities, or weight discordancy.

Table 5 shows the risk factors predictive of neurodevelopmental delay determined using regression analysis. The results were calculated after adjusting for confounding factors such as gestational age at delivery, birth weight, placental weight, NICU admission, intubation, and ventilator use. In DC twins, the male sex had an odds ratio of 2.628 (CI: 0.951–7.261, *p* = 0.062); the presence of BPD, 25.645 (CI: 7.690–85.527, *p* = 0.000); and RDS, 3.708 (CI: 1.080–12.725, *p* = 0.037). Moreover, weight discordancy was confirmed as a predictor of neurodevelopmental delay, with an odds ratio of 6.805 (CI: 2.563–18.064, *p* = 0.000).

## 4. Discussion

Birth weight discordancy was related to obstetrical complications and poor neonatal outcomes, and it can be a risk factor for neurodevelopmental delay in twin pregnancies.

In a twin pregnancy, the uterus, which can provide an adequate blood supply to one fetus, has to sustain two fetuses. In the process of adapting to this, maternal complications and fetal problems such as FGR arise [17]. Intertwin growth discordancy is one of the known factors influencing the prognosis of twin pregnancy. In a twin pregnancy, the reason for the discordancy should be considered in the context of chorionicity. In MC twins, the inequality of placental sharing between fetuses is the main reason for discordancy, along with vascular anastomosis and the malposition of the cord insertion site [8,18]. In DC twins, discordancy may be due to the difference in genetic growth potential between the two fetuses or the suboptimal invasion of the trophoblast on one side during implantation [6].

Discordancy can influence maternal obstetrical complications and neonatal outcomes; however, the prognosis according to chorionicity has only been studied to a limited extent. Breathnach et al. reported that the discordancy of MC twins had a worse prognosis at every level of discordance compared with DC twins [10]. However, Coutinho et al. found that in discordant twins, chorionicity was an independent factor for neonatal outcomes [12,13]. In the present study, we grouped the participants according to chorionicity (MC vs. DC) and compared the neonatal outcomes of the discordant and concordant twins in each group. The relationship between discordant large and small twins and prognosis was also evaluated. In the present study, 19.0% of MC pregnancies and 20.8% of DC pregnancies were discordant, which were higher than the average known incidence of 10–15%, despite excluding TTTS [8,9]. Pregnancy achieved via ART was significantly higher in DC pregnancies than in MC ones; discordant twins were more common as well. Similar to a previous study, the incidence of preeclampsia was high in DC discordant twins, with no statistical significance in relation to the incidence of preeclampsia in MC twins [19]. The incidence of iatrogenic delivery was higher in discordant twins than in concordant twins, regardless of chorionicity. Therefore, if twin discordancy is suspected, the time of delivery should be ascertained by closely monitoring the condition of the mother and fetus until delivery. Among MC twins, discordant twins had a lower birth weight and a higher neonatal mortality rate than concordant twins, but they did not significantly differ in the 1- and 5-min Apgar scores. This is because TTTS and MCMA twins were excluded. Among DC twins, discordant twins had lower birth weight, frequent hospitalizations, and increased morbidity than concordant twins. The developmental delay between 1 and 2 years of age was also significantly higher in discordant twins. According to Table 2, the neurodevelopmental delay was significantly higher in discordant DCDA twins than in concordant DCDA twins, whereas in MCDA twins, no significant difference was noted. We assumed that discordancy is related to neurodevelopmental delay in any type of twins because there was a difference in the percentage (6.5% vs. 13.6%); however, in this data, the number of MCDA twins (especially discordant MCDA twins) was too small. When comparing large and small MC discordant twins, no statistically significant differences were found when comparing outcomes such as birth weight, neonatal mortality, neonatal morbidity, and developmental delay. However, among DC discordant twins, large babies had a higher incidence of male sex and were hospitalized more frequently, but there was no difference in neonatal morbidity between the two groups. This validates the findings of a previous study [13], which also found no significant difference in the prognosis of larger and smaller babies in discordant twins regardless of chorionicity. Among MC twins, when comparing concordant and discordant large babies, no significant differences were observed in birth weight and prognosis (i.e., neonatal morbidity and mortality). Discordant small neonates had significantly lower birth weight and higher neonatal mortality rates than concordant twins, but no difference was observed in the incidence of morbidity. Therefore, larger neonates in discordant twins have growth similar to concordant twins. According to Lewi et al., the weight difference is likely due to the FGR of discordant small neonates [6]. Among DC twins, males were more common in discordant large neonates than in concordant neonates, but no difference was observed in neonatal morbidity or mortality. However, discordant large neonates had a high incidence of PVL and developmental delay. After the regression analysis, discordancy was found to be a plausible risk factor for developmental delay. The strengths of this study were as follows: (1) the data were from a single tertiary center and a single race (Asian); and (2) the study carefully compared the outcomes of discordant twins according to chorionicity. However, this study has several limitations. First, the number of participants was small. Further research with a larger sample population is needed. Second, there was a selection bias in this study. Third, in evaluating the neurodevelopmental delay, conducting a long-term follow-up until at least 2 years of age is recommended.

Many studies have tackled the relationship between discordancy and subsequent neurodevelopment, but the results have been inconsistent [20,21,22]. Halling et al. assessed neurodevelopment in terms of recognition, language, and motor skills in discordant small and large twins after 3 years of age and reported that small babies were more disadvantaged [23]. Groene et al. also demonstrated a difference in long-term neurodevelopmental outcomes between smaller and larger twins at birth in discordant MC twins [24]. However, because our study excluded TTTS and analyzed neurodevelopmental delay after 1 year of age, it would not be appropriate to make a direct comparison with previous studies and deem our findings inconsistent. In the present study, when infants with and without developmental delay were compared according to chorionicity, the developmental delay was observed in 7.8% and 5% of MC and DC twins, respectively. However, if TTTS is not excluded, the frequency is thought to be higher. Hack et al. reported that a mild neurodevelopmental delay 2 years after birth was significantly increased in MC twins compared with that in DC twins [20]. Overall, the risk of neurodevelopmental delay increased in the presence of PTB, lower body weight, or morbidities. In DC twins, discordancy was found to increase this risk by approximately 6.8 times; however, in MC twins, discordancy was a significant risk factor for the neurodevelopmental delay. In DC twins, weight discordancy and PTB can be risk factors for the neurodevelopmental delay.

## 5. Conclusions

Birth weight discordancy was related to obstetrical complications and poor neonatal outcomes, and it can be a risk factor for neurodevelopmental delay in twin pregnancies.

## Figures and Tables

**Figure 1 medicina-59-00493-f001:**
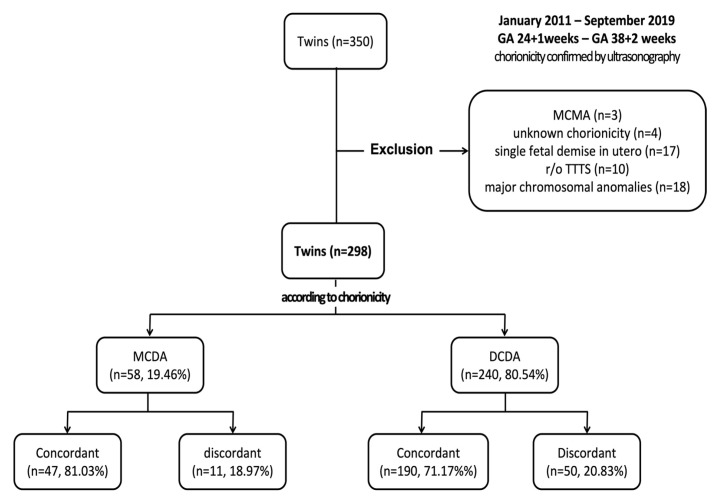
Enrollment process flowchart. DCDA, dichorionic diamniotic; GA, gestational age; MCDA, monochorionic diamniotic; MCMA, monochorionic monoamniotic; TTTS, twin-to-twin transfusion syndrome.

**Table 1 medicina-59-00493-t001:** Maternal characteristics and obstetrical complications in concordant and discordant twins according to chorionicity.

	Sets of Twins (*n* = 298)			MCDA Set of Twins (*n* = 58)			DCDA Set of Twins (*n* = 240)		
	Concordant (*n* = 237)	Discordant (*n* = 61)	*p*-Value	Concordant (*n* = 47)	Discordant (*n* = 11)	*p*-Value	Concordant (*n* = 190)	Discordant (*n* = 50)	*p*-Value
Age (year)	31.94 ± 4.08	33.07 ± 4.92	**0.021 ***	30.28 ± 3.75	32.55 ± 3.58	**0.011 ***	32.35 ± 4.06	33.18 ± 5.17	0.138
Nulliparous	161 (67.9%)	45 (73.8%)	0.057	32 (68.1%)	6 (54.6%)	0.340	129 (67.9%)	39 (78.0%)	**0.028 ***
Pre-pregnancy BMI	22.12 ± 3.74	22.67 ± 4.00	0.158	21.61 ± 3.40	20.80 ± 3.04	0.310	22.24 ± 3.82	23.10 ± 4.08	0.053
ART	119 (52.2%)	38 (64.4%)	**0.025 ***	7 (15.9%)	1 (11.1%)	0.875	112 (60.9%)	37 (74.0%)	**0.028 ***
Pre-eclampsia	18 (7.6%)	12 (19.7%)	**<0.001 ***	2 (4.3%)	2 (18.2%)	0.064	16 (8.5%)	10 (20.0%)	**0.002 ***
Gestational diabetes	34 (14.4%)	10 (16.4%)	0.684	2 (4.3%)	1 (9.1%)	0.699	32 (16.9%)	9 (18.0%)	0.918
Placenta previa	4 (1.7%)	3 (4.9%)	**0.025 ***	0(0.0%)	1 (9.1%)	**0.041 ***	4 (2.1%)	2 (4.0%)	0.055
Threatened preterm	146 (61.6%)	31 (50.8%)	**0.039 ***	32 (68.1%)	4 (36.4%)	**0.012 ***	114 (60.0%)	27 (54.0%)	0.332
PROM	57(24.1%)	15 (24.6%)	0.996	15 (31.9%)	2 (18.2%)	0.311	42 (22.1%)	13 (26.0%)	0.490
Postpartum bleeding	6 (2.5%)	2 (3.3%)	0.893	0 (0.0%)	1 (9.1%)	**0.041 ***	6 (3.2%)	1 (2.0%)	0.775
Cause of delivery Elective Spontaneous Iatrogenic	98 (44.3%) 103 (46.6%) 20 (9.1%)	12 (18.5%) 34 (52.3%) 19 (29.2%)	**<0.001 ***	18 (38.3%) 27 (57.5%) 2 (4.3%)	3 (27.3%) 5 (45.5%) 3 (27.3%)	**0.002 ***	79 (41.6%) 90 (47.4%) 21 (11.1%)	9 (18.0%) 26 (52.0%) 15 (30.0%)	**<0.001 ***

ART, artificial reproductive technique; BMI, body mass index; PROM, premature rupture of amniotic membrane, * *p*-values of <0.05 are shown in bold with an asterisk (*).

**Table 2 medicina-59-00493-t002:** Neonatal outcomes of concordant and discordant twins according to chorionicity.

	Each Twin Baby (*n* = 596)			MCDA Each Twin Baby (*n* = 116)			DCDA Each Twin Baby (*n* = 480)		
	Concordant (*n* = 474)	Discordant (*n* = 122)	*p*-Value	Concordant (*n* = 94)	Discordant (*n* = 22)	*p*-Value	Concordant (*n* = 380)	Discordant (*n* = 100)	*p*-Value
Gestational age at delivery (weeks)	35.07 ± 2.51	34.41 ± 3.10	0.030 *	34.81 ± 2.53	33.20 ± 5.30	0.178	35.10 ± 2.64	34.67 ± 2.32	0.140
Sex, male	253 (53.4%)	62 (50.8%)	0.687	53 (56.4%)	11 (50.0%)	0.761	201 (52.9%)	51 (51.0%)	0.822
Birth weight (g)	2246.24 ± 478.42	1996.02 ± 603.81	<0.001 *	2194.89 ± 471.05	1802.50 ± 779.22	0.032 *	2258.94 ± 479.99	2038.60 ± 553.76	<0.001 *
Mean discordancy (%)	8.43	29.95		13.15	28.30		8.40	30.18	
FGR, *n*(%)	61 (12.9%)	50 (41.0%)		9 (9.6%)	10 (45.5%)		52 (13.7%)	40 (40%)	
1-min Apgar score (<7)	99 (20.9%)	36 (29.5%)	0.056	22 (23.4%)	7 (31.8%)	0.584	77 (20.3%)	29 (29.0%)	0.098
5-min Apgar score (<7)	8 (1.7%)	7 (5.7%)	0.026 *	2 (2.1%)	2 (9.1%)	0.336	6 (1.6%)	5 (5.0%)	0.097
NICU admission	312 (66.1%)	108 (88.5%)	<0.001 *	61 (68.8%)	19 (86.4%)	0.165	248 (65.4%)	89 (89.0%)	<0.001 *
Neonatal death	5 (1.1%)	5 (4.1%)	0.053	0 (0.0%)	3 (13.6%)	0.004 *	5 (1.3%)	2 (2.0%)	0.971
Composite morbidity	127 (26.8%)	45 (36.9%)	0.037 *	25 (26.6%)	7 (31.8%)	0.819	102 (26.8%)	38 (38.0%)	0.039 *
Intubation	56 (11.9%)	24 (19.7%)	0.036 *	15 (16.1%)	6 (27.3%)	0.363	41 (10.9%)	18 (18.0%)	0.078
Ventilator use (nasal cPAP)	158 (33.5%)	52 (42.6%)	0.075	36 (38.7%)	9 (40.9%)	1.000	122 (32.2%)	43 (43.0%)	0.057
O2 supply	205 (43.4%)	62 (50.8%)	0.174	47 (50.5%)	10 (45.5%)	0.848	158 (41.7%)	52 (52.0%)	0.083
Neurodevelopmental delay	17 (3.6%)	16 (13.1%)	<0.001 *	6 (6.5%)	3 (13.6%)	0.492	11 (2.9%)	13 (13.0%)	<0.001 *

FGR, fetal growth restriction; NICU, neonatal intensive care unit; PVL, periventricular leukomalacia, * *p*-values of <0.05 are shown with an asterisk (*).

**Table 3 medicina-59-00493-t003:** Comparison of neonatal outcomes of concordant twins with discordant larger and smaller neonates according to chorionicity.

	MCDA Each Twin Baby (*n* = 116)						DCDA Each Twin Baby (*n* = 480)					
	Concordant (*n* = 94)	Discordant Large (*n* = 11)	Discordant Small (*n* = 11)	*p*- Value ^α^	*p*- Value ^β^	*p*- Value ^ɤ^	Concordant (*n* = 380)	Discordant Large (*n* = 50)	Discordant Small (*n* = 50)	*p*-Value ^α^	*p*-Value ^β^	*p*-Value ^ɤ^
GA at delivery (wks)	34.81 ± 2.53	33.20 ± 5.43	33.20 ± 5.43	0.353	0.353		35.10 ± 2.64	34.67± 2.33	34.67 ± 2.33	0.275	0.217	
Sex, male	53 (56.4%)	5 (45.5%)	5 (45.5%)	0.712	0.712	1.000	201 (52.9%)	36 (72.0%)	15 (30.0%)	0.016 *	0.004 *	<0.001 *
Birth weight (g)	2194.89 ± 471.05	2087.73 ± 850.55	1517.27 ± 610.48	0.689	<0.001 *	0.086	2258.94 ± 479.99	2385.40 ± 458.66	1691.80 ± 404.66	0.079	<0.001 *	<0.001 *
FGR, *n*	9	0	10				52	1	39			
1-min AS (<7)	22 (23.4%)	4 (36.4%)	3 (27.3%)	0.567	1.000	1.000	77 (20.3%)	13 (26.0%)	16 (32.0%)	0.452	0.087	0.659
5-min AS (<7)	2 (2.1%)	1 (9.1%)	1 (9.1%)	0.722	0.722	1.000	6 (1.6%)	2 (4.0%)	3 (6.0%)	0.526	0.127	1.000
NICU admission	64 (68.1%)	9 (81.8%)	10 (90.9%)	0.555	0.222	1.000	248 (65.4%)	39 (78.0%)	50 (100.0%)	0.106	<0.001 *	0.001 *
Neonatal death	0 (0.0%)	1 (9.1%)	2 (18.2%)	0.195	0.003 *	1.000	5 (1.3%)	0 (0.0%)	2 (4.0%)	0.908	0.417	0.475
Composite morbidity	25 (26.6%)	3 (27.3%)	4(36.4%)	1.000	0.742	1.000	102 (26.8%)	17 (34.0%)	21 (42.0%)	0.371	0.039 *	0.537
Developmental delay	6 (6.4%)	2 (1 8.2%)	1 (9.1%)	0.427	0.742	1.000	11 (2.9%)	9 (18.0%)	4 (8.0%)	<0.001	0.151	0.234

FGR, fetal growth restriction; GA, gestational age at delivery; NICU, neonatal intensive care unit; PVL, periventricular leukomalacia, * *p*-values of <0.05 are shown with an asterisk (*), *p*-value ^α^, between concordant and discordant large; *p*-value ^β^, between concordant and discordant small; *p*-value ^ɤ^, between discordant large and discordant small.

**Table 4 medicina-59-00493-t004:** Neonatal outcomes in relation to neurodevelopmental delay according to chorionicity.

	Each Twin Baby (*n* = 596)			MCDA Each Twin Baby (*n* = 116)			DCDA Each Twin Baby (*n* = 480)		
	No Developmental Delay (*n* = 563)	Developmental Delay (*n* = 33)	*p*-Value	No Developmental Delay (*n* = 107)	Developmental Delay (*n* = 9)	*p*-Value	No Developmental Delay (*n* = 456)	Developmental Delay (*n* = 24)	*p*-Value
GA at delivery (weeks)	35.09 ± 2.49	32.27 ± 3.79	<0.001 *	34.77 ± 3.00	31.32 ± 4.73	0.061	35.16 ± 2.35	32.63 ± 3.42	0.001 *
Birth weight(g)	2218.67 ± 500.36	1791.67 ± 613.99	<0.001 *	2167.20 ± 529.50	1565.00 ± 652.04	0.002 *	2230.74 ± 493.11	1876.67 ± 590.75	0.001 *
Placental weight (g)	1042.35 ± 203.78	929.38 ± 227.14	0.003 *	1003.55 ± 206.04	808.89 ± 132.61	0.006 *	1052.10 ± 202.00	976.52 ± 240.92	0.084
Sex, male	291 (51.7%)	24 (72.7%)	0.030 *	58 (54.2%)	6 (66.7%)	0.709	233 (51.1%)	18 (75.0%)	0.038 *
NICU admission	389 (69.1%)	31 (93.9%)	0.004 *	74 (69.2%)	9 (100.0%)	0.113	315 (69.1%)	22 (91.7%)	0.033 *
1-min AS (<7)	119 (21.1%)	16 (48.5%)	0.001 *	23 (21.5%)	6 (66.7%)	0.009 *	96 (21.1%)	10 (41.7%)	0.034 *
5-min AS (<7)	12 (2.1%)	3 (9.1%)	0.056	4 (3.7%)	0 (0.0%)	1.000	8 (1.8%)	3 (12.5%)	0.006 *
Composite morbidity	151 (26.8%)	21 (63.6%)	<0.001 *	27 (5.2%)	5 (55.6%)	0.117	124 (27.2%)	16 (66.7%)	<0.001 *
Placental infection	32 (6.7%)	4 (12.1%)	0.404	3 (2.8%)	1 (11.1%)	0.322	29 (6.4%)	3 (12.5%)	0.620
Weight discordance	106 (18.8%)	16 (48.5%)	<0.001 *	19 (17.8%)	3 (33.3%)	0.483	87 (19.1%)	13 (54.2%)	<0.001 *

NICU, neonatal intensive care unit; weight discordance, >20% of weight difference, * *p*-values of <0.05 are shown with an asterisk (*).

**Table 5 medicina-59-00493-t005:** Multiple regression analysis for the risk factors predictive of neurodevelopmental delay 1 year after birth.

	Each Twin Baby (*n* = 596)				DCDA Each Twin Baby (*n* = 480)			
	Odds Ratio	95% CI		Significance	Odds Ratio	95% CI		Significance
		Lower Limit	Upper Limit			Lower Limit	Upper Limit	
Sex, male	2.711	1.115	6.590	0.028 *	2.628	0.951	7.261	0.062
BPD	6.191	1.526	25.121	0.011 *	25.645	7.690	85.527	0.000 *
IVH	7.438	1.094	50.568	0.040 *	5.133	0.978	26.952	0.053
RDS	4.965	2.057	11.986	0.000 *	3.708	1.080	12.725	0.037 *
Sepsis	2.360	0.996	5.590	0.051	2.476	0.962	6.377	0.060
Presence of discordance	3.461	1.503	7.969	0.004 *	6.805	2.563	18.064	0.000 *

BPD, bronchopulmonary dysplasia; IVH, intraventricular hemorrhage; RDS, respiratory distress syndrome; * *p*-values of <0.05 are shown with an asterisk (*).

## Data Availability

The datasets used and/or analyzed during the current study are available from the corresponding author upon reasonable request.

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
