# Peer review of "Perinatal Outcomes and Neurodevelopment 1 Year after Birth in Discordant Twins According to Chorionicity"

_medicina, 2023, doi:10.3390/medicina59030493_

Round 1

Reviewer 1 Report

Authors want the perinatal outcomes and neurodevelopment 1 year after birth in 2 discordant twins according to chorionicity. It is a nice endeavour.

In discordant twins, chorionicity has been reported as an independent factor for ne- 64 onatal outcomes [12,13]. Same is your finding in this study. Any new thing added?

The primary aim of this study is to compare the maternal outcome of discordant,

Whether you are looking for maternal outcome (Line 70)/ or neonatal outcome after 1 year in various group of babies?

Discussion: Please do provide the important findings of your analysis in first few lines of discussion. Then start discussing about the findings and your concepts.

Table 4

Neurodevelopmental delay, is a broad terminology.

Which group of baby developing what kind developmental delay ( gross motor, fine motor, speech or psycho-social) should have been correlated for better understanding.

Reviewer 2 Report

THIS PARAGRAPH NEED TO BE NOT AT THE END OF THE DISCUSSION ,BUT DURING THE DISCUSSION

Strengths of this study include: (1) the data is all from a single tertiary center and a 346 single race (Asian); (2) the study carefully compares the outcomes of discordant twins 347 according to chorionicity. However, this study had several limitations. First, the number 348 of participants was small. Further research with a larger sample population is needed. 349 Second, the hospital where the study was conducted was a tertiary hospital and an inten-350 sive care center for high-risk pregnant women. So, there was a selection bias in this study. 351 Third, in evaluating the neurodevelopmental delay, conducting a long-term follow up 352 until at least 2 years of age is recommended.

FURTHERMORE: PLEASE ERASE THIS SENTENCE ,IT IS NORMALE THAT THIS TYPE OF PREGNANCY NEED TO BE FOLLOWED IN ATERATIARRY REFRALL CENTER ( Second, the hospital where the study was conducted was a tertiary hospital and an inten-350 sive care center for high-risk pregnant women HERMORE.)

COULD CLARIFY IN SPONTANEUS DELIVERY OF set of MCDT AT WHICH WEEKS OF GESTATION WERE BORNED?WHY NOT ELECTIVE CESAREAN SECTION? PLEASE COULD CLARIFY?

POST PARTUM HM 1000CC OF BLEEDING AS DEFINITION IN SPOTANEUS DELIVERY ,WHILE FOR CESAREAN SECTION?

PULMUNARY PROFIPHILAXIS WAS PERFROMED IN ALL PATIENTS?

THERE ARE PARTICULARY SITUATION DURING THE SPONTANEUS DELIVERY AFTER THE FIRST TWIN WAS BORNED? VACUM EXTRATION ON THE SECOND TWIN?OBSTETRICS MANEUVRES?
